

# Palatability and pharmacokinetics of flunixin when administered to sheep through feed

Danila Marini[1,2], Joe Pippia[3], Ian G. Colditz[2], Geoff N. Hinch[1], Carol J. Petherick[4] and Caroline Lee[2]

[1] School of Environmental and Rural Science, University of New England, Armidale, New South Wales, Australia
[2] Agriculture, Commonwealth Scientific and Industrial Research Organisation, Armidale, New South Wales, Australia
[3] Pia Pharma Pty Ltd, Gladesville, New South Wales, Australia
[4] Queensland Alliance for Agriculture and Food Innovation, The University of Queensland, Brisbane, Queensland, Australia

Corresponding author
Danila Marini,
danila.marini@csiro.au

## ABSTRACT

Applying analgesics to feed is a potentially easy method of providing pain-relief to sheep and lambs that undergo painful husbandry procedures. To be effective, the medicated feed needs to be readily accepted by sheep and its consumption needs to result in therapeutic concentrations of the drug. In the present experiment, pelleted feed was supplemented with flunixin (4.0 mg/kg live weight) and offered to eight sheep. To test the palatability of flunixin, the individually penned sheep were offered normal feed and feed supplemented with flunixin in separate troughs for two consecutive days. A trend for a day by feed-type (control versus flunixin supplemented) interaction suggested that sheep may have had an initial mild aversion to pellets supplemented with flunixin on the first day of exposure, however, by on the second day there was no difference in consumption of normal feed and feed supplemented with flunixin. To test pharmacokinetics, sheep were offered 800 g of flunixin supplemented feed for a 12 h period. Blood samples were taken over 48 h and plasma drug concentrations were determined using ultra-high-pressure liquid chromatography, negative electrospray ionisation and tandem mass spectrometry. The mean $\pm$ S.D. time required to reach maximum concentration was $6.00 \pm 4.14$ h and ranged from 1 to 12 h. Average maximum plasma concentration was $1.78 \pm 0.48$ µg/mL and ranged from 1.61 to 2.80 µg/mL. The average half-life of flunixin was $7.95 \pm 0.77$ h and there was a mean residence time of $13.62 \pm 1.17$ h. Free access to flunixin supplemented feed enabled all sheep to obtain inferred therapeutic concentrations of flunixin in plasma within 6 h of starting to consume the feed. Provision of an analgesic in feed may be an alternative practical method for providing pain relief to sheep.

Subjects Agricultural Science, Veterinary Medicine
Keywords Flunixin, Pharmacokinetics, Sheep, Oral administration, Pain relief, Non-steroidal anti-inflammatory, Palatability

## INTRODUCTION

Flunixin meglumine is a potent non-steroidal anti-inflammatory drug (NSAID) that is commonly used in veterinary medicine for its anti-inflammatory, analgesic and antipyretic

properties. Like other NSAIDs, flunixin reduces inflammation by inhibiting cyclooxygenase and, in turn, decreasing the production of prostaglandins (*Cheng, Nolan & McKellar, 1998*), which are important inflammatory mediators. Flunixin is known to be effective at relieving pain in various domesticated species such as horses (*Keegan et al., 2008*; *Toutain et al., 1994*) and cattle (*Currah, Hendrick & Stookey, 2009*) and is currently registered for use for these animals in the USA, Europe and Australia. Although flunixin has also been shown to be effective for pain relief in sheep (*Paull et al., 2007*; *Welsh & Nolan, 1995*), there are currently no NSAIDs registered in Australia for use in sheep. Pain relief can be impractical and costly to administer to livestock raised in extensive systems due to the necessity for repeated application over time and the limited availability of registered drugs (*Lizarraga & Chambers, 2012*). A potential practical method of providing pain relief is through oral administration, allowing farmers to either provide NSAIDs as a drench or through feed in the form of granules or a liquid formulation. It is known that the rumen can decrease the bioavailability of NSAIDs if they are administered orally (*Mosher et al., 2012*; *Odensvik, 1995*). In previous work to counteract the reduced bioavailability when administering NSAIDs orally to cattle the dose given was double compared with that recommended for parenteral administration (*Coetzee et al., 2012*).

Incorporation of flunixin to an animal's diet could possibly elicit a neophobic response or reduced feed intake if flunixin is unpalatable. Therefore the objectives of this study were (1) to test the palatability of flunixin and (2) determine the pharmacokinetics of flunixin in sheep plasma when feed supplemented with flunixin was offered. We hypothesised that all sheep would achieve therapeutic concentrations of flunixin in plasma when consuming feed supplemented with flunixin.

## MATERIALS AND METHODS

### Experimental animals

Nine, 2-year-old, maiden Merino ewes with mean live weight of $38.8 \pm 2.83$ kg (mean $\pm$ S.D.) were used in this study. Sheep were clinically healthy at the time of the study. Upon entry to the experiment the animals body condition was checked, they were then vaccinated with Glanvac® 6S B12 (Zoetis Animal Health, Silverwater, NSW, Australia) and drenched with Firstmectin (Virbac, Milperra, NSW, Australia), Flukazole C (Virbac, Milperra, NSW, Australia) and Rycozole (Novartis, North Ryde, NSW, Australia) at the manufacturers' recommended dose rates. Following vaccination and drenching the sheep were then monitored daily for any signs of ill health, such as behavioural and respiratory changes. There was a month between drenching treatments and the pharmacokinetic experiment. The sheep were housed in individual pens in a covered shed which was open on the North face and were in close proximity to allow visual and social interaction with other experimental animals. Animals were fed a complete pelleted ration (Ridley Agriproducts, Australia; 17% crude protein dry matter; 9.04 MJ/kg dry matter). During acclimation, sheep were offered a small excess of feed over their previous day's intake (between 800– 1,000 g) supplemented with 100 g of oaten chaff daily so that some residual feed was left at the end of each day. Water was also provided ad-libitum. The experiment

was undertaken at CSIRO's FD McMaster Laboratory, Armidale, New South Wales (NSW). The protocol and conduct of the experiment was approved by The CSIRO Armidale Animal Ethics Committee under the NSW Animal Research Act, 1985 (ARA 14/01).

## Palatability test

One week prior to the start of the experiment, each animal was acclimatised to eating from two troughs within its pen and daily feed intake was monitored. The palatability test ran over 2 days; in the morning sheep were offered feed in two troughs, one containing 2 kg of the standard pelleted ration and the other containing 2 kg of the same standard pelleted diet supplemented with 20 mL (300 mg) of liquid flunixin (Flunixin Oral solution, 15 mg/mL; Pia Pharma Pty Ltd, Gladesville, NSW, Australia). The amount of flunixin added per kg of feed was equivalent to an approximate single dose for the live weight of each ewes (i.e., eating 1 kg of the supplemented feed would deliver 1 dose at 4 mg/kg body weight). The feed was prepared each morning by putting the liquid flunixin onto the pellets and thoroughly mixing them together in the trough; even incorporation of the liquid was characterised by the change in colour of the pellets. Following flunixin application the trough did not appear to be wet and there was no free liquid present at the bottom of the trough. Both troughs were placed into the pen simultaneously and the location of the trough containing flunixin supplemented feed was alternated for the second day of testing.

## Pharmacokinetic protocol

After the palatability test, the ewes were kept in a paddock for a 2-week flush-out period. They were then returned to the same individual pens that were used for the palatability test, 1 week prior to the beginning of the pharmacokinetic experiment. The sheep were again fed the complete pelleted ration ad libitum supplemented with 100 g of oaten chaff once a day. The day prior to supplementation of feed with flunixin, sheep were weighed and had the wool clipped from their necks. To allow for intensive blood sampling, catheters were inserted aseptically in the left jugular vein using a 12 G catheter needle to puncture the vein. A piece of catheter tubing was then threaded through the needle and then, to ensure the catheter was inserted correctly, the line was flushed with heparinised saline and then liquid withdrawn until blood was seen flowing. Catheters were then re-flushed with heparinised saline. The catheter needle was removed and the line was sealed with a three-way tap adaptor containing a luer lock syringe port. The line was secured to the animal at the exit point with Elastoplast tape, the remaining catheter tubing was then encased in 7.5 cm wide Elastoplast bandage which was gently wrapped around the sheep's neck.

On the day of the study, sheep were offered 800 g of feed containing a dose of flunixin (at a rate of 4.0 mg/kg live weight) adjusted for each animal's body weight. Flunixin was added to feed as described for the palatability test. The first sheep was presented with the flunixin supplemented feed at 0700 h and the remaining sheep were given their feed at 2 min intervals thereafter. Blood samples (10 mL) were collected before the flunixin supplemented feed was offered (0 h) and at 5, 10, 15, 20, 30, 45 min and 1, 2, 4, 6, 8, 12, 24, 36, 48 h relative to the time each sheep was first observed to have consumed some of the supplemented feed. Prior to the collection of each blood sample, 2 mL of blood was

withdrawn from the catheter and discarded to ensure that fresh blood was collected. Blood samples were centrifuged (2,000× g) and the separated plasma collected and frozen at −20 °C. Residual feed remaining in the trough was weighed at each blood sampling time point until 12 h post-initial ingestion.

## Plasma flunixin concentration determination

Plasma samples were transported frozen to Pia Pharma Pty Ltd, Gladesville, NSW for flunixin concentration determination using ultra-high-pressure liquid chromatography, negative electrospray ionisation and tandem mass spectrometry (UHPLC/−ve ESI MS/MS). Each plasma sample was thawed to room temperature on the day of analysis. For determination, a 250 µL aliquot of each plasma sample was dispensed into a 2 mL polypropylene centrifuge tube. Flunixin-d3 internal standard (50 µL of 2.0 µg/mL flunixin-d3) was added and the sample mixed gently prior to the addition of 350 µL acetonitrile. The sample was vortexed (1 min) and centrifuged (13,000 rpm/5 min) to remove any sediment. Type 1 water for UHPLC applications (0.5 mL) was then added to the extract and the mixture was filtered through a 0.45 µm filter prior to determination. An aliquot of sample extract (5 µL) was injected into an Eksigent® Ekspert™ ultraLC 100-XL Liquid Chromatograph fitted with a Supelco Ascentis® Express 50 × 2.1 mm, 2.7 µm analytical column maintained at 40 °C. A gradient elution program, based on a combination of 0.1% formic acid and acetonitrile as mobile phase constituents operating at 0.4 mL min$^{-1}$, resolved flunixin and flunixin-d3 (retention time of 2.5 min) from matrix interferences and endogenous sample components. The identity of peaks was predicted using an AB Sciex API 3200 triple-quadrupole mass spectrometer interfaced with the liquid chromatograph. The detector was configured with a proprietary turbo V source for desolvation and operated in negative electrospray ionisation mode (−4,500 V), desolvation temperature 550 °C, for optimum analyte selectivity and sensitivity. The transitions for flunixin and flunixin-d3 were 295.1 → 191.0 and 298.2 → 254.0 respectively.

Matrix matched calibration standard solutions of flunixin were prepared at increasing concentrations between 10 and 4,000 ng/mL in plasma from animals prior to treatment. The calibration curve was prepared by plotting the nominal flunixin concentration ($x$ axis) against the determined peak area ratio of flunixin and flunxin-d3 for each calibrator. A correlation co-efficient ($r$) greater than 0.99 was required for the calibration curve to be used for quantitative purposes. Analyte concentrations were calculated using the peak area ratio of flunixin detected in each sample relative to the corresponding flunixin-d3 internal standard, and the regression equation of the calibration curve.

Method accuracy and precision were monitored with the inclusion of fortified quality control samples. Four plasma samples containing flunixin concentrations of 13.1, 328.5, 1314.1, 3942.3 ng/mL ($n = 3$) were prepared on the day of the analysis. The mean percentage of accuracy was 90.8% at lower limit of quantification (LLOQ) and 102.9–111.6% at all other concentrations. The coefficient of variation at LLOQ was 2.9%, and 1.3–3.1% at other concentrations. Quality control data were acceptable.

**Table 1** Palatability test results (mean ± S.D.) for the effect of interaction of feed type (flunixin supplemented or control) by day (1 or 2) and location (left or right) on feed intake (g) in eight sheep.

| Location | Day 1 | | Day 2 | |
| --- | --- | --- | --- | --- |
| | Control | Flunixin supplemented | Control | Flunixin supplemented |
| Left | 906.00 ± 426.28 | 451.75 ± 338.76 | 707.88 ± 451.40 | 590.67 ± 518.79 |
| Right | 1158.88 ± 330.73 | 562.83 ± 358.93 | 364.50 ± 446.95 | 561.63 ± 309.77 |
| Mean | 1050.50 ± 365.42 | 499.36 ± 348.68[a] | 560.71 ± 449.62[a] | 574.07 ± 414.19 |

**Notes.**

[a] Mean is significantly different to the control feed on day 1 ($P < 0.05$).

## Statistics

Palatability data were analysed with R-Project (https://www.r-project.org/) using nlme package (*Pinheiro et al., 2015*) to perform a linear mixed model analysis. Fixed effects included feed type (flunixin present or absent), day (1 or 2), and location of flunixin supplemented feed trough (left or right) and the interaction of feed type by day. Sheep number was fitted as a random effect. Results are presented as mean ± S.D. Data were tested for normality using the Shapiro–Wilk test. $P < 0.05$ was considered as statistically significant.

## Pharmacokinetic analysis

Pharmacokinetic modelling of flunixin in plasma was performed using an open source pharmacokinetic program (PK Solver; China Pharmaceutical University, Nanjing, Jiangsu, China) (*Zang et al., 2010*). Using non-compartmental analysis, the maximum flunixin concentration (Cmax) in plasma, the time required to reach Cmax (Tmax), mean residence time (MRT) and elimination half-life (t1/2) were determined for each animal. The area under the concentration vs. time curve (AUC0–t) was calculated using the linear trapezoidal rule. Pharmacokinetic parameters were estimated for each animal from which mean values ± S.D. were calculated.

## RESULTS

### Palatability

One ewe was excluded from data analysis as she did not consume any of the feed containing flunixin on either day. Location of the different feeds (left or right trough) had no effect ($P = 0.81$) on the amount of each feed (flunixin supplemented versus control) that was consumed. Although there was no main effect of feed type across days ($P = 0.10$), a trend was observed for the day by feed type interaction ($P = 0.08$). On day 1, animals consumed on average 551.14 ± 446.68 g more of the control feed than the flunixin supplemented feed ($P = 0.02$). Whereas on day 2 there were no differences observed in the consumption of control feed and feed supplemented with flunixin ($P = 0.95$). On day 2, consumption of control feed decreased on average by 489.79 ± 468.53 from the quantity consumed on day 1 ($P = 0.03$). Consumption of feed supplemented with flunixin was comparable on days 1 and 2 ($P = 0.73$, Table 1).

**Table 2  Variability in feed intake of eight sheep that were offered 800 g of flunixin supplemented feed for a 12 h period.**

| Time feed was weighed (h) | Average intake (g) ± S.D. | Median (g) | Range (g) |
|---|---|---|---|
| 0.08 | 174.69 ± 112.12 | 205.75 | 21.50–357.50 |
| 0.17 | 26.25 ± 29.66 | 18.50 | 0.00–71.00 |
| 0.25 | 18.63 ± 16.01 | 15.00 | 0.00–48.00 |
| 0.33 | 6.06 ± 12.27 | 1.00 | 0.00–36.00 |
| 0.50 | 23.44 ± 17.30 | 25.00 | 0.00–50.50 |
| 0.75 | 13.13 ± 20.79 | 6.50 | 0.00–62.50 |
| 1 | 6.19 ± 15.72 | 0.00 | 0.00–45.00 |
| 2 | 91.38 ± 58.90 | 75.25 | 32.00–211.00 |
| 4 | 151.63 ± 39.12 | 148.25 | 89.00–220.00 |
| 6 | 141.56 ± 56.38 | 149.75 | 68.00–211.00 |
| 8 | 88.38 ± 59.54 | 71.25 | 31.00–194.00 |
| 12 | 58.69 ± 114.15 | 8.50 | 0.00–332.50 |

## Pharmacokinetics

The sheep took between 8 and 12 h to consume the total 800 g of flunixin supplemented feed on offer. Most of the sheep spread meals throughout the day except for ewe 466 who ate 350 g of feed in the first 5 min and ewe 627 who consumed 332.5 g in the last 4 h of the 12 h period. Flunixin was absorbed rapidly, all sheep had detectable plasma concentrations (>20 ng/mL) at 10 min after initial consumption of supplemented feed with the exception of one animal (ewe 627), who only ate 21.5 g of feed in the first 10 min.

All sheep started to eat within a few minutes after the provision of feed. There was large variability between sheep in the amount of feed that was consumed at each time-point (Table 2). When animals had free access to feed, the majority of sheep (7 out of 8) achieved plasma flunixin concentrations above 1.0 μg/mL within 2 h of starting to consume the supplemented feed, with maximum concentrations (between 1.33 and 2.80 μg/mL) being reached on average by 6 h. Flunixin concentration time curve (mean ± S.D.) in all sheep plasma over a period of 48 h is shown in Fig. 1. This led to a large variability in the Tmax, which ranged from 1 to 12 h. The Cmax average was 1.78 ± 0.48 μg/mL and the flunixin meglumine plasma $t_{1/2}$ was 7.95 ± 2.19 h (Table 3).

## DISCUSSION

Concentrations measured in this study were somewhat lower compared with those reported in our previous study (*Marini et al., 2015*) where flunixin concentration in plasma reached values between 2.6 and 4.1 μg/mL 2 h after a single oral dose (4 mg/kg) in sheep. Reports of therapeutic concentrations of flunixin in farm animals are limited, however, *Toutain et al. (1994)* reported therapeutic effects in horses when plasma concentrations reached 0.2–0.9 μg/mL. The results of the current study suggest that the plasma flunixin concentrations achieved following consumption of supplemented feed may be within the therapeutic range for sheep.

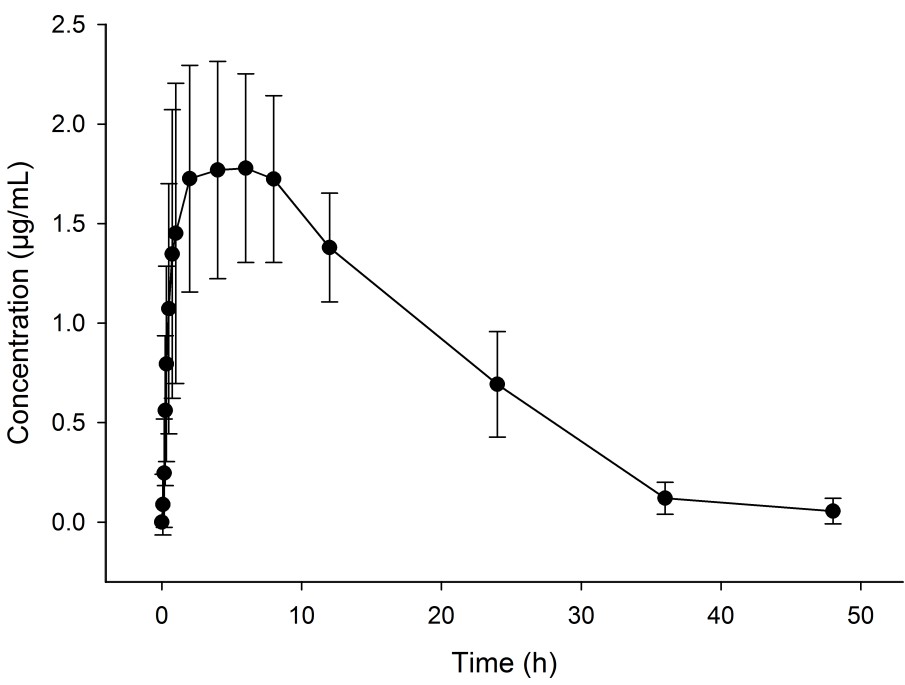

**Figure 1** Flunixin in plasma concentration time curve (means ± S.D.) of eight sheep over a 48 h period following administration of flunixin (4.0 mg/kg) through pelleted feed.

**Table 3** Flunixin pharmacokinetic parameters following oral administration through 800 g of pelleted feed to eight sheep at a dose rate of 4 mg/kg.

| Parameter (units) | Sheep ID | | | | | | | | |
|---|---|---|---|---|---|---|---|---|---|
| | 305 | 466 | 580 | 612 | 621 | 627 | 648 | 732 | Mean ± S.D. |
| $t_{1/2}$, h | 4.59 | 5.39 | 8.23 | 6.29 | 7.31 | 4.85 | 11.04 | 5.19 | 7.95 ± 2.19 |
| $T$max, h | 8.00 | 1.00 | 6.00 | 6.00 | 2.00 | 12.00 | 12.00 | 4.00 | 6.00 ± 4.14 |
| $C$max, µg/mL | 2.39 | 1.61 | 2.18 | 1.89 | 2.16 | 1.33 | 1.63 | 2.80 | 1.78 ± 0.48 |
| AUC0-$t$, µg/mL*h | 29.96 | 38.00 | 38.21 | 40.99 | 42.78 | 31.84 | 42.75 | 36.05 | 37.68 ± 4.77 |
| MRT, h | 9.36 | 14.34 | 13.36 | 13.43 | 12.98 | 15.80 | 19.48 | 9.32 | 13.59 ± 3.31 |

**Notes.**
[a]Flunixin non-compartmental pharmacokinetics (PK Solver; China Pharmaceutical University, Nanjing, Jiangsu, China) (*Zang et al., 2010*).
$t_{1/2}$, elimination half-life; $c$max, the maximum flunixin concentration in plasma; $T$max, the time required to reach $C$max; AUC0-$t$, area under the concentration vs. time curve; MRT, mean residence time.

Although displaying an initial (day 1) preference for control pelleted feed over flunixin-supplemented feed, there was no overall feed preference effect observed. The initial preference of control pelleted feed may have been due to the novelty of the odour or flavour of flunixin. Odour and flavour help sheep distinguish different types of feed and they are more likely to eat novel feeds that contains some familiar flavours (*Hinch et al., 2004*; *Launchbaugh, Provenza & Werkmeister, 1997*). Sheep are known to avoid novel feed types for several days before they start to consume it (*Chapple, Wodzicka-Tomaszewska & Lynch, 1987*). Adding flunixin to a feed with which the ewes were familiar, may have reduced any neophobia. With the exception of one ewe who did not consume any feed

supplemented with flunixin over the two days, the intake of supplemented and control feeds was similar on the second day of testing.

The pharmacokinetics of flunixin has been investigated following intramuscular and intravenous administration (*Cheng, McKeller & Nolan, 1998*; *Welsh, McKellar & Nolan, 1993*). When administered intravenously, the elimination half-life of flunixin meglumine has been reported to be 2.48 h (*Cheng, McKeller & Nolan, 1998*) and 3.83 h (*Welsh, McKellar & Nolan, 1993*). The elimination half-life observed in the current study (following oral administration) was longer (7.95 ± 2.19 h). Differences were also observed for the MRT of flunixin following intravenous versus oral administration, with MRT in plasma being 3.20 ± 0.18 h (*Cheng, McKeller & Nolan, 1998*) compared with 13.59 ± 3.31 h in the current study. When flunixin is administered intramuscularly and intravenously it is typically given as a bolus dose, which permits a uniform pattern of absorption and elimination to occur. The longer half-life and mean retention time observed in this study is likely due to animals consuming their dose of flunixin over an extended period of time, rather than as a bolus. The AUC observed in the current study (37.62 ± 4.77 μg/mL*h) was similar to that reported by *Cheng, McKeller & Nolan (1998)* (30.61 ± 3.41 μg/mL*h). It is probable that our higher AUC was due to the higher dose rate used in our study.

The pharmacokinetics of orally administered flunixin has been studied in goats (*Königsson et al., 2003*), horses (*Pellegrini-Masini, Poppenga & Sweeney, 2004*; *Welsh et al., 1992*) and cattle (*Odensvik, 1995*). Following oral administration of a bolus dose in the absence of feed in these species, flunixin is absorbed rapidly and concentrations can still be detected up to 30 h after administration (*Königsson et al., 2003*; *Odensvik, 1995*). Horses that had ad libitum access to hay following the oral administration of flunixin had a slower absorption of flunixin and a lower Cmax although concentrations of flunixin in plasma were maintained for longer when animals had access to feed compared with when they were fasted (*Welsh et al., 1992*). The AUC was not significantly different when animals were fasted or non-fasted, suggesting that the absorption of flunixin is not affected by the presence of feed. In the current study, flunixin was found to be absorbed rapidly when consumed with feed, with detectable levels present within 10 min in sheep that consumed more than 22 g. Flunixin concentrations remained detectable, but were below inferred therapeutic concentrations, for 36–40 h after consumption of the flunixin supplemented feed ceased (Fig. 1). Currently there are no toxicity data for flunixin in sheep, however the animals used in this study did not show any visible side effects as a result of consuming flunixin supplemented feed.

Previous work in cattle by *Odensvik (1995)* showed that oral administration of flunixin (2.2 mg/kg) as a granule inhibited the production of prostaglandin $PGF_{2}\alpha$ by up to 60%, which was as effective as the standard therapeutic dose of flunixin (2.2 mg/kg) used parenterally. Although the authors did not directly measure the effectiveness of oral flunixin at reducing inflammation, they concluded that an anti-inflammatory effect was likely to occur due to reduced production of $PGF_{2}\alpha$ which acts as a pro-inflammatory factor following injury (*Ricciotti & FitzGerald, 2011*). Although further studies are required it is expected that oral administration of flunixin could provide effective pain relief in sheep.

In conclusion, results of this study demonstrate that when flunixin is administered orally through feed to sheep it is absorbed rapidly into the bloodstream and despite variability in consumption rates of feed, all sheep reached inferred therapeutics concentrations of flunixin within 6 h of starting to consume the feed. Further studies are required to investigate potential binding of flunixin to various feed components and potential impacts that such binding may have on toxicity. The possible mild aversion to feed supplemented with flunixin on day 1 did not persist on day 2 indicating that the medicated feed is readily accepted by sheep. Supplementation of feed with flunixin may provide a practical way to provide pain relief to sheep prior to and after painful husbandry procedures thus eliminating the need for multiple injections, reducing handling stress and minimising labour requirements.

## ACKNOWLEDGEMENTS

We thank Sue Belson, Brad Hine, Dominic Niemeyer, Tim Dyall, Aurélie Bussy and Etienne Goumand for their assistance during the experiment. We also thank Paul Mills (The University of Queensland) for his advice on pharmacokinetic analyses.

### Funding

This work was funded by CSIRO; the University of New England; and funding contributors, Meat and Livestock Australia; and the Commonwealth Government. The funders had no role in study design, data collection and analysis, decision to publish, or preparation of the manuscript.

### Grant Disclosures

The following grant information was disclosed by the authors:
CSIRO.
University of New England.
Meat and Livestock Australia.
Commonwealth Government.

### Competing Interests

The authors declare there are no competing interests.

### Author Contributions

- Danila Marini conceived and designed the experiments, performed the experiments, analyzed the data, wrote the paper, prepared figures and/or tables.
- Joe Pippia contributed reagents/materials/analysis tools, wrote the paper.
- Ian G. Colditz conceived and designed the experiments, performed the experiments, reviewed drafts of the paper.
- Geoff N. Hinch, Carol J. Petherick and Caroline Lee conceived and designed the experiments, reviewed drafts of the paper.

## Animal Ethics

The following information was supplied relating to ethical approvals (i.e., approving body and any reference numbers):

CSIRO Armidale Animal Ethics Committee (ARA 14/01).

## Data Availability

The raw data is available at CSIRO: 10.4225/08/56CEA5FBD5584.

## Supplemental Information

Supplemental information for this article can be found online at http://dx.doi.org/10.7717/peerj.1800#supplemental-information.

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
