# Peer review of "Palatability and pharmacokinetics of flunixin when administered to sheep through feed"

_PeerJ, doi:10.7717/peerj.1800_

## Round 0.1 · original submission · Major Revisions

Both reviewers have provided extensive comments (and see the annotated manuscript from Reviewer 1). Please consider all their suggestions in the revised manuscript.

Reviewer 1 ·

Basic reporting

The article provides useful information on highlighting the effects of flunixin dietary supplementation on feed intake and plasma pharmacokinetics in sheep. It gives new insights into this field. However, there are a lot of grammar, stylistic and syntax errors. In some cases, these errors negatively influence the understanding of the text. I strongly advise the authors to consult an expert in the use of English scientific language before publishing their manuscript.

Experimental design

Number of animals is small. However, I think that this is a preliminary study.

Validity of the findings

In general, experiment is correctly implemented.
It is not necessary to present raw data in the manuscript.
At the same time, authors sometimes reached to contradictory conclusions (i.e. neophobia results - conclusions).

Additional comments

Please check corrections and comments in the attached pdf file

Annotated reviews are not available for download in order to protect the identity of reviewers who chose to remain anonymous.

Reviewer 2 ·

Basic reporting

Rewording and editing of some parts of the manuscript are suggested. For details, please see under "General Comments for the Author".

Table headings and figure legends should be descriptive enough so that the reader can understand them without needing to go back to look for information in the main text.

Please present data as mean ± SD or median and range as appropriate.

For symbols denoting significant differences (e.g. * in Figure 1), please make sure to indicate what groups were compared.

Experimental design

Please indicate how 1) the PK parameters were computed, 2) the distribution of individual PK parameters were analysed and 3) the results were presented (mean ± SD; median and range).

Please provide the correct amount of flunixin added to the feed. If 20 mL of a solution containing 15 mg of flunixin per mL, then it should be 300 mg instead of 200 mg. Please clarify this issue.

Please explain how it was assured that the full amount of flunixin was impregnated in the feed and not left in the mixing container or the trough.

Validity of the findings

The authors mentioned that “animals were acclimatised to eating from two troughs and daily food intake was recorded”. However, the results from the acclimatisation period are not reported in the Results section of the manuscript. This is important information to know whether the sheep had a preference to eating from a particular trough or not before commencement of the palatability trial.

Additional comments

This manuscript investigated the acceptance of flunixin-medicated feed and its plasma pharmacokinetics in sheep. The paper is interesting and clinically useful, however I have some concerns. The presentation of the obtained results can be improved and re-editing would enhance the readability of the manuscript. For more specific details see below.


Title

Consider changing the title to “Palatability and pharmacokinetics of flunixin when administered to sheep through feed”


Abstract

Lines 24-26. Consider changing to “It is also important to determine if therapeutic drug concentrations can be achieved when administering a medicated feed to sheep”.

Lines 28-29. Change “Ultra High Pressure Liquid Chromatography” for “ultra-high-pressure liquid chromatography-tandem mass spectrometry”.

Lines 34-37. Please rephrase this sentence since pain relief was not assessed and sheep, but not lambs, were used in the study. Therefore, these two topics should not be included as part of the conclusion in this study.

Within the Abstract, please provide information on the palatability study.


Introduction

Line 44. Include “drug” after “anti-inflammatory”.

Line 47. Change “prostaglandin (Cheng et al. 1998b), an important inflammatory mediator.” for “prostaglandins (Cheng et al. 1998b), which are important inflammatory mediators.”

Line 64. Delete was from “We hypothesised was that…”.


Materials and Methods

Line 69. Where the sheep clinically healthy at the time of the study and if so, how this was assessed?

Lines 79-80. The authors mentioned that “animals were acclimatised to eating from two troughs and daily food intake was recorded”. However, the results from the acclimatisation period are not reported in the Results section of the manuscript. This is important information to know whether the sheep had a preference to eating from a particular trough or not before commencement of the palatability trial.

Line 83. Please provide the correct amount of flunixin added to the feed. If 20 mL of a solution containing 15 mg of flunixin per mL, then it should be 300 mg instead of 200 mg. Please clarify this issue.

Lines 86-88. How was it assured that the full amount of flunixin was impregnated in the feed and not left in the mixing container or the trough?

Lines 96-97. Please indicate if the left or right jugular vein was used. If there was not consistency on using a particular side, then use “a jugular vein” instead of “the jugular vein”.

Line 109. Please be more specific as to “was observed consuming the medicated feed.”

Lines 110-11. Please be more specific in the amount (in mL) of blood withdrawn and discarded.

Lines 116-118. Please change “Ultra High Liquid Chromatography Tandem Mass
Spectrometry (UHPLC-MSMS)” for “ultra-high-pressure liquid chromatography-tandem mass spectrometry (UHPLC-MS/MS)”

Line 123. Please specify the type of water used.

Line 131. Delete was from “…quadrupole mass spectrometer was interfaced”

Line 133. The abbreviation “(-ve ESI)” is not needed in the manuscript.

Line 147. Define LLOQ –(lower limit of quantification).

Lines 154-155. This sentence should be moved to the Results section of the manuscript.

Lines 160-162. Please indicate how the PK parameters were computed.

Lines 163-164. Please indicate how the distribution of individual PK parameters was analysed and indicate how the results were presented (mean ± SD; median and range).


Results

Lines 167-168. Please provide P value for the comparison stated in this sentence.

Lines 172-173. Please provide P value for the comparison stated in this sentence.

Lines 167-173. Results from the acclimatisation period should be presented here. Please also present and compare all the results from the palatability trial in a table instead of only presenting the overall feed consumption as in Figure 1. This way the readers will have access to the results according to what it was mentioned in the M&M section of the manuscript: feed type (flunixin present or absent), day (1 or 2), location of medicated feed trough (left or right) and the interaction of feed type by day.

Lines 175-176. This sentence could be reworded and better placed under M&M. Pharmacokinetic parameters were tabulated for each animal and reported as mean and SD.

Lines 176-177. This sentence could be reworded as “Mean ± SD plasma concentration versus time curves following oral dosing was plotted for flunixin (Fig. 1)

Line 180. Please report Cmax values without rounding up to higher decimal value.

Line 181. Consider changing “…800 g of feed.” for “…800 g of flunixin-impregnated feed.”

Line 183. If still taking about the PK trial, there is no need to use “of the first day” since this trial took place over 12 hours within the same day. Please rephrase the last part of the sentence “…and ewe 627 who consumed 332.5 g in the last 4 h of the first day.”


Discussion

Lines 188-190. This should be part of the Results section.

Lines 196-197. Since no pharmacodynamic trials were carried out in this study, this sentence should be rephrased to suggest that plasma flunixin concentrations achieved in this study might have been within therapeutic range in sheep.

Lines 215-217. Please remove or rephrase this sentence. A direct comparison of flunixin’s MRT in plasma (in the current study) and exudate and transudate (Chang et al. 1998a) cannot be made since it is well known that flunixin and many other NSAIDs accumulate in inflamed tissue.

Lines 208-219 and 220-232. The same information is provided in these two paragraphs and is almost verbatim from each other.

Lines 244-246. Please change “…, but were below therapeutic concentrations,” for “…, but were below inferred therapeutic concentrations,”.

Lines 262-265. The study was carried out in sheep and the obtained results may not be relevant for lambs. Also, consider discussing further the practicality (pros and cons) of administering flunixin-medicated feed to sheep raised in commercial extensive systems.


References

Lines 279-281, 289-291, 299-301 and 326-327. Use “sentence case” instead of “title case” for the titles in these references.

Lines 311-313. Please correct the title of this reference.


Table and Figures

Headings and legends should describe the information such that the reader can understand them without needing to go back to look for information in the main text.

Please present data as mean ± SD or median and range as appropriate.

For symbols denoting significant differences (e.g. * in Figure 1), please make sure to indicate what groups were compared.

---

## Round 0.2 · Major Revisions

Your manuscript has been re-reviewed by the 2 original reviews and both have suggested additional edits. Please modify according the suggestions given.

Reviewer 1 ·

Basic reporting

The article includes sufficient introduction and background to demonstrate how the work fits into the broader field of knowledge.

Experimental design

The submission clearly defines the research question and methods are described with sufficient information to be reproducible by another investigator.

Validity of the findings

The data were statistically sound and controlled. In most cases, the conclusions were appropriately stated and limited to those supported by the results.

Additional comments

Please check comments in the attached file

Annotated reviews are not available for download in order to protect the identity of reviewers who chose to remain anonymous.

Reviewer 2 ·

Basic reporting

Please see under "General Comments for the Author"

Experimental design

Please see under "General Comments for the Author"

Validity of the findings

Please see under "General Comments for the Author"

Additional comments

I think that the manuscript has improved from the previous version. I personally think that some sections still need some rewording.

Abstract
Although information on the palatability study was provided, the major findings of this trial are still missing in the abstract. Please amend it for completeness.

Results
Line 186: Move “(Table 1)” to the end of the paragraph.

Discussion
Line 273: Please eliminate “lambs” from this sentence since the reported study was undertaken in ewes only.

Tables and Figures
Table 1
I believe that a more logical way of presenting the data, and that also follows what is written in the results section (lines: 182-190), is by having “Day 1” and “Day 2” as headings in columns 1 and 2 and “Flunixin supplemented” and “Control” underneath those headings. In the same Table 1, it would also be beneficial to present P values for the comparisons made.
The heading should include the animal species used and the number of individuals from which data were collected.

Figure 1
There is no need for keeping Figure 1 since it only partially illustrates what it could be presented in an amended Table 1.

Figure 2
Please provide the number of sheep from which data were obtained.

General comment
Both SE and SD are used throughout the manuscript. For consistency, only one of them should be used. Unless there is an editorial preference, I would suggest to provide SD instead of SE.

---

## Round 0.3 · accepted · Accept

Thank you for improving your manuscript.